# Artificial Neural Network as a Tool to Predict Facial Nerve Palsy in Parotid Gland Surgery for Benign Tumors

**DOI:** 10.3390/medsci8040042

**Published:** 2020-10-07

**Authors:** Carlos M Chiesa-Estomba, Jon A Sistiaga-Suarez, José Ángel González-García, Ekhiñe Larruscain, Giovanni Cammaroto, Miguel Mayo-Yánez, Jerome R Lechien, Christian Calvo-Henríquez, Xabier Altuna, Alfonso Medela

**Affiliations:** 1Department of Otorhinolaryngology, Osakidetza, Donostia University Hospital, 20014 San Sebastian, Spain; jasistiaga@osakidetza.eus (J.A.S.-S.); tirititet@gmail.com (J.Á.G.-G.); ekhinel@gmail.com (E.L.); XABIER.ALTUNAMARIEZCURRENA@osakidetza.eus (X.A.); 2Biodonostia Health Research Institute, 20014 San Sebastian, Spain; 3Head & Neck Study Group of Young-Otolaryngologists of the International Federations of Oto-rhino-laryngological Societies (YO-IFOS), 13005 Marseille, France; giovanni.cammaroto@hotmail.com (G.C.); miguelmmy@gmail.com (M.M.-Y.); lechienj@gmail.com (J.R.L.); Christian.calvo.henriquez@gmail.com (C.C.-H.); 4Department of Otolaryngology—Head & Neck Surgery, Morgagni Pierantoni Hospital, 47121 Forli, Italy; 5Otorhinolaryngology—Head and Neck Surgery Department, Complexo Hospitalario Universitario A Coruña (CHUAC), 15006 A Coruña, Spain; 6Department of Human Anatomy & Experimental Oncology, University of Mons, 7000 Mons, Belgium; 7Department of Otolaryngology—Hospital Complex of Santiago de Compostela, 15704 Santiago de Compostela, Spain; 8LEGIT Health, 48071 Bilbao, Spain; alfonso@legit.health

**Keywords:** artificial neural network, parotid, surgery, facial nerve, palsy

## Abstract

(1) Background: Despite the increasing use of intraoperative facial nerve monitoring during parotid gland surgery or the improvement in the preoperative radiological assessment, facial nerve injury (FNI) continues to be the most feared complication; (2) Methods: patients who underwent parotid gland surgery for benign tumors between June 2010 and June 2019 were included in this study aiming to make a proof of concept about the reliability of an artificial neural networks (AAN) algorithm for prediction of FNI and compared with a multivariate linear regression (MLR); (3) Results: Concerning prediction accuracy and performance, the ANN achieved the highest sensitivity (86.53% vs 46.23%), specificity (95.67% vs 92.59%), PPV (87.28% vs 66.94%), NPV (95.68% vs 83.37%), ROC–AUC (0.960 vs 0.769) and accuracy (93.42 vs 80.42) than MLR; and (4) Conclusions: ANN prediction models can be useful for otolaryngologists—head and neck surgeons—and patients to provide evidence-based predictions about the risk of FNI. As an advantage, the possibility to develop a calculator using clinical, radiological and histological or cytological information can improve our ability to generate patients counselling before surgery.

## 1. Introduction

Despite the increasing use of intraoperative facial nerve monitoring during parotid gland surgery (PGS), well-known anatomic landmarks and improvement in the preoperative radiological assessments, facial nerve injury (FNI) continues to be the most severe complication after PGS. Transient facial nerve dysfunction occurs in 20–65% of patients undergoing a parotidectomy, whereas permanent, definitive facial nerve palsy occurs in 0–7% of those patients; [1,2,3,4] impairing significantly patients’ quality of life [5,6].

Machine learning (ML) is a subset of artificial intelligence (AI) that enables computers to learn from historical data, gather insights and make predictions about new data using the model learned, with an increasing use related to medical application in recent years [7].

Artificial neural networks (ANNs) represent an innovative subfield of ML inspired by the human brain, which is capable of learning and accurately solving complicated relationships between input and modeled output data. Structurally, ANN comprises an input, hidden and output layers of multiple layers of interconnected nodes, in which each node performs a series of nonlinear calculations based on its inputs and signals from other nodes connected to it. Each connection—much like the synapses in the human brain—transmits data from one node to the next [7]. These characteristics made ANN a useful choice for predictive inferences that can be used to provide support for clinical decision-making, for classification purposes or to establish prognosis [8,9,10,11].

Recently, different studies have been published about the use of ANNs related to the medical and the otolaryngological field and to estimate prognosis in some tumors [8,9,10,11,12,13,14]. However, the use of ANNs specifically to evaluate the risk of FNI after parotid gland surgery for benign tumors has not been previously studied. The authors hypothesize that ANN will improve prediction of patients at risk, being the objective of this study to evaluate the effectiveness of the use of ANN in prognostication of facial palsy in this subset of patients.

## 2. Materials and Methods

After approval from the Ethics Committee of our Center, a retrospective study was conducted which included a group of patients underwent parotid gland surgery for benign tumors between June 2010 and June 2019 aiming to make a proof of concept about the reliability of ANN for prediction of FNI. Case identification was made through a review of our department’s databases using the International disease classification (ICD-9–10). Inclusion criteria for study correspond to patients with at least 18 years of age with a clinically and radiologically evident benign tumor in the parotid gland. Patients were excluded if they were treated non-surgically, in case of tumor affecting the accessory lobe (V), in case of revision surgery or if the final histology corresponds to malignancy.

### 2.1. Prognostic Parameters

The patients’ medical histories were analyzed to obtain information about demographic data (age, sex), clinical presentation, preoperative assessments, radiological test (CT, MRI), diagnosis and surgical management. For variable selection, the study used the results from previously published data and expert knowledge. Finally, the correlation between the clinical variable and FNI outcome can be evaluated through the ANN analysis. (Figure 1).

The predictors included for analysis were age, sex, tumor volume or size (anterior to posterior, mid to lateral and cephalic to caudal) and tumor size (<3 cm or >3 cm) measured on MRI or CT scans and then confirmed on final histology after surgery, type of resection and areas resected according to the ESGS classifications [15,16]. Only benign histologies were considered in this study (pleomorphic adenoma, Warthin tumor, oncocytoma, etc.). The primary outcome of interest was the presence or of transient of permanent FNI in at least one branch of the facial nerve after surgery.

### 2.2. Surgical Technique

Before surgery, each patient underwent a Ultrasound-guided fine-needle aspiration to establish the nature of the tumor (benign or malignant) and had a computerized tomography or magnetic resonance study. Parotidectomy was generally performed using modified Blair incision or facelift incision, depending on patient preference. A monopolar electric scalpel was used to raise skin flap, make the dissection of the anterior edge of the sternocleidomastoid muscle and the posterior bottom of the digastric muscle. After this, we created a tunnel in the pre-tragal area until the cartilaginous, pointer was found. Once the main trunk of the facial nerve was identified, the parotid tissue was divided using and harmonic scalpel (Harmonic Focus, Ethicon Endo-Surgery, Inc., Cincinnati, OH, USA) or bipolar cautery. The type and extension of the resection of the parotid gland were defined in keeping with the parotidectomy classification of the European Salivary Gland Society (ESGS) [15,16]. In all cases, the facial nerve was monitored and stimulated before and after resection. At the end of the surgery, we routinely attached a vacuum suction drain (Jost–Redon).

Facial nerve function was systematically assessed immediately after surgery and the day after surgery after asking the patient to furrow their brow, close their eyes with force, pucker the lips into a whistling shape and show their teeth. The facial function was measured during follow-up according to the House–Brackman scale.

### 2.3. Statistical Analysis

The quantitative variables within the study are expressed as a mean ± standard deviation; the results are expressed as both total and percentage. Differences among groups with FNI and those without were analyzed using the Shapiro–Wilk test. The difference was considered statistically significant if the P value was less than 0.05. Statistics were calculated using JASP (Version 0.11.1. University of Amsterdam, Amsterdam, The Netherlands) (https://jasp-stats.org/).

### 2.4. Model Training

First, a predictive model based on those predictors that were collected in the clinic was developed over the training data. A supervised learning method was used in this study. Our dataset was split using an 80:20 stratified sampling according to the FNI outcome whereby the machine-learning algorithms were trained using 80% of the available cases and tested using the remaining 20%. Continuous variables were normalized and categorical variables label encoded with no additional preprocessing. An artificial neural network with a specific architecture for tabular data were chosen, with two hidden layers of 200 and 100 neurons, respectively. The ANN embeds the categorical variables and applies a dropout before feeding input data into the linear layers and applies batch normalization to continuous variables. The output layer was a simple two-neuron layer in which each neuron is one of the target categories. The model was trained for 20 epochs with a learning rate of 0.01 and another four epochs with a learning rate ten times smaller (Figure 1).

### 2.5. Validation

Classification performance of the machine-learning algorithms was then evaluated comparing area under the receiver operating characteristic curve (AUC-ROC) for internal validation. All predictive models were then externally validated reporting sensitivity, specificity, positive predictive value (PPV), negative predictive value (NPV) and accuracy. Additionally, we compared the performance of this ANN model with logistic regression model (LRM). ANN was developed and performed with PyTorch version 1.6 (https://pytorch.org/), MLR was developed and performed with Sciket-Learn version 0.17.1 (http://scikit-image.org). Data preprocessing and analysis was performed with Pandas version 1.1.0 (https://pandas.pydata.org).

## 3. Results

### 3.1. Demographic Data

During the study period, 356 patients were operated because of benign tumors in the parotid gland. Of these, 345 patients could be included in the final analysis due to missing information from 11 patients. Of these, 192 (55.7%) were male and 153 (44.3%) were female. The age average was 58 years old. (SD: 14 = min: 18/max: 87). Fifty-one patients (14.8%) were diabetic and 111 (32.2%) were hypertensive. The mean follow-up was 11 months (min: 6/max: 24) (Table 1). There were no differences between both groups (*p* = 0.947).

Regarding the final histological diagnosis, the most common was pleomorphic adenoma in 153 (44.3%) of the cases. The most common type of resection was the type I (141; 40.9%). Anatomically, the parotid tail was the most common sublocation involved (162; 47%) and about levels resected, the most common was the level II (159; 46.1%) followed by the combination of level I and II (120; 34.8%). Eighty-four (24.4%) patients presented a transient facial nerve paresis and 12 (3.5%) a definitive facial nerve or facial nerve branch paralysis, being the marginal mandibular nerve the most common transient or definitively branch affected (48; 13.8%). (Table 1).

### 3.2. Artificial Neural Network Vs Multivariate Logistic Regression Results

In relation to the accuracy and performance prediction, the ANN achieve a highest sensitivity, specificity, PPV, NPV, ROC–AUC and accuracy than LRM (Table 2 and Figure 2). Overall, the algorithms are based towards the majority class of non-FNI patients, showing low sensitivity and PPV corresponding to the high number of false negatives. Looking for the most influential predictors in the performance of the ANN models, the situation of the tumor on the gland (mid, superior and deep lobe), the volume of the tumor in the anterior-posterior axis and cephalo-caudal axis, the histology (pleomorphic adenoma), the age and the type of resection were the most significant factors related with the risk of FNI (Table 3). In the LRM, the type of resection, situation, sex and age were the most weighted variables, being consider significant for FNI (Table 3).

## 4. Discussion

In this study, researchers explored for the first time the performance of an ANN to predict FNI after surgery for benign tumor of the parotid gland, including clinical variables that surgeons can obtain during the clinical exploration, diagnosis workup and including retrospective data related to the surgery that can be inferred in the clinical scenario.

There are different factors described that can increase the risk of temporary or permanent FNI after parotid gland surgery for benign tumors like the old age, malignancy, tumor size (>70 cm^3^), operative time, the need for revision surgery due to recurrence, tumor subsite location (superficial vs deep) and extent of surgery [1,17,18,19,20,21,22,23,24,25,26,27]. However, the data reported are heterogeneous, due to the absence of a common language to indicate extent or type of resection and the different types of techniques described like extracapsular dissection, partial parotid gland resection, superficial or total parotidectomy. For this reason, authors included their results according to the ESGS classification system [15,16].

According to the performance of our ANN algorithm, (1) there is a significant risk of FNI in patients whose tumor is located in the mid-portion of the gland over the main trunk of the facial nerve or in the most superior part over the frontal or orbital branch of the facial nerve; (2) the type of histology in case of pleomorphic adenoma, maybe due to the high risk of recurrence pushes surgeons into a better resection trying to avoid recurrence; (3) the volume of the tumor in the cephalo-caudal and antero-posterior axis due to the need for a higher facial branch dissection; (4) the age probably related to the slower recovery after FNI in older patients compared with younger patients and (5) the need for an extended resection compared with lest extended (Table 3). Regarding LRM, type of resection, situation, sex and age were the most weighted factors related to the risk of FNI. In contrast, in both models, the size of the tumor (>3 cm) was not considered by both methods a relevant predictor (Table 3).

As mentioned above, ANNs are biologically inspired computer programs designed to simulate the way in which the human brain processes and interpret information. The algorithm gathers information, and then used their knowledge by detecting patterns or relationships from these data to learn through its own experience and not from programming [8]. Structurally, an ANN is formed by hundreds of single units, the so-call artificial neurons or perceptron’s, connected with coefficients, which constitute the neural structure and are organized in layers.

In the ANN algorithm, the power of neural computations comes from connecting perceptron’s in a network. Each perceptron has weighted inputs, a transfer function and one output. The behavior of a neural network is determined by the transfer functions of its neurons, by the learning rule establish and by the architecture itself. Each weight corresponds to an adjustable parameter working as a parameterized system. Moreover, wants to be the weighed sum of the inputs, who allows the activation of each perceptron. After the activation signal, this is passed through a transfer function to produce a single output of the perceptron [28].

Here we can highlight an advantage of the algorithm, because when we run the ANN the activation function wants to confer nonlinearity to the architecture improving their capacity to learn any complex relationship between input and output data. Then, during training, the inter-unit connections wants to be optimized until the error in predictions is minimized and the network reaches the specified level of accuracy. Finally, once the network is trained and tested it can be given new input information to predict a specifical output.

Historically, regression models have been commonly used in medicine. Despite the great quality of this algorithm to enhance diagnostic and management accuracy, this have two main shortcomings including the assumption of normality for residuals beside their inability to identify nonlinear relationships [29]. Machine-learning tools such as ANN methods are evolving to avoid limitations of traditional outcome prediction methods, gaining increased applications in the field of otolaryngology [30]. However, the utilization of these methods is scarce likely due to a lack of wide understanding or easily implementable application.

Previous articles have investigated the application of ANN in otolaryngology—head and neck surgery. Abouzari et al. compare an ANN model with a logistic regression model to predict the risk of vestibular schwannoma recurrence, obtaining a higher sensitivity and specificity with the use of the ANN [13]. Alabi et al. published a study in which they summarize data from Finland and Brazil to estimate the risk of locoregional recurrence in early stage SCC of the oral tongue. Here, the authors compared the use of an ANN versus logistic regression (overall accuracy was 92.7% vs 86.5%) using a web-based application [14].

However, to translate findings from ML or ANN to the clinical environment, we need to understand the differences between the architecture of ANN algorithms and the classical statistics models. ANN focuses on how all the variables interrelate among them, taking this information to make predictions about an unknown variable [31]. Meanwhile, statistics is primarily focused on making inferences: analyzing how components relate to one another through the development of a statistical model [32,33]. Thus, both fields overlap substantially providing complementary results.

This study has several limitations. First, this study aimed to build and test prediction models to discriminate patients at risk of FNI optimally and not a casual model, hence the rank of each variable does not necessarily show the importance of that variable in the chain of causation due to the black–box effect, to decrease the risk of bias we try to estimate the importance of each variable inside the ANN according to our needs for prediction. The retrospective nature of our data collection, the exclusion of revision surgery, parotid accessory lobe tumor or malignant histology for this first approach, the fact that four surgeons perform almost all the surgeries can be consider limitations from our study, being necessary to include surgeries from different teams to evaluate the performance of our algorithms in different surgical environments. Another limitation of our study is the small sample size, have not been large enough to appropriately train and validate our ANN model. In this way, future studies may benefit from large national databases which can provide significantly larger cohorts for developing a more precise and reproducible algorithm. Finally, the needs to consider all those possible residual measured or unmeasured confounders that could have influenced the outcomes. In addition, these results need to be interpreted with precautions, because as we say above, this is a proof of concept looking for future application of ANN in Otolaryngology-Head and Neck Surgery.

As a future perspective, comparison with other ML algorithms wants to be performed. In addition, a web-based application including the algorithm of our ANN was designed to test the ability to predict of our model in a clinical setting intended to improve the prediction ability through an increasing input of anonymized clinical data from different clinical centers around the world. Aiming to develop a FNI calculator using clinical, radiological and histological or cytological information to improve our ability to generate patients counselling before surgery and be aware of the most feared complication, looking for a better surgical outcome.

## 5. Conclusions

This study demonstrates that the use of ANN prediction models can be useful for otolaryngologists—head and neck surgeons and patients to provide evidence-based predictions about the risk of FNI after PGS for benign tumors. The constructed ANN model was superior to logistic regression in predicting FNI with higher sensitivity, specificity, accuracy, PPV and NPV. However, prospective studies advocated to evaluate the real possibilities of this technique are necessary to validate this proof-of-concept in the clinical practice.

## Figures and Tables

**Figure 1 medsci-08-00042-f001:**
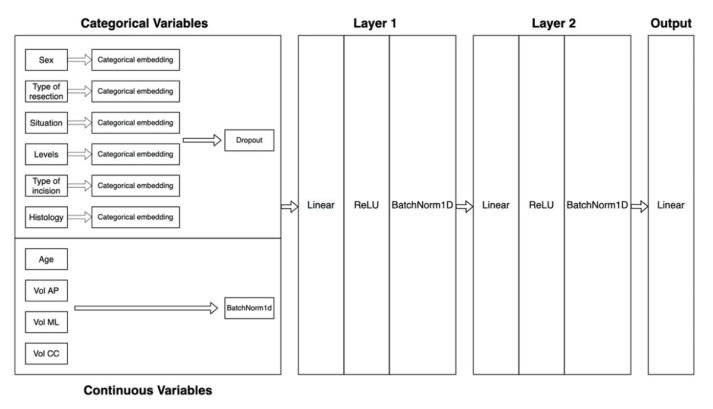
Artificial neural network architecture.

**Figure 2 medsci-08-00042-f002:**
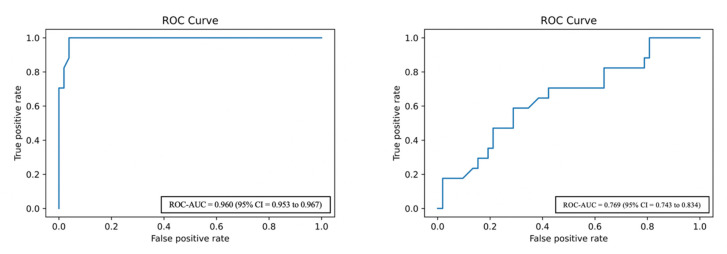
ROC–AUC curve. (**left**) Artificial neural network ROC–AUC curve results; (**right**) multivariate logistic regression ROC–AUC curve results.

**Table 1 medsci-08-00042-t001:** Demographic and clinical data.

Variable	N	%
Sex		
Male	192	55.7
Female	153	44.3
Mean Age	58 ± 13.9 years (min: 18/max: 87)	
Size		
<3 cm	216	62.6
>3 cm	129	37.4
Maximum length per plane		
- Anterior to posterior	2.45 ± 1.15 cc (min: 0.9/max: 6.5)	
- Medial to lateral	2.17 ± 1.07 cc (min: 0.7/max: 6)	
- Cephalic to caudal	2.22 ± 0.99 cc (min: 0.76/max: 6.8)	
Type of resection		
I	141	40.9
II	48	13.9
III	99	28.7
IV	57	16.5
Anatomic situation in the gland		
Parotid tail	162	47
Mid lobe	45	13
Superior lobe	21	6.1
Deep lobe	24	7
Superior and middle lobe	15	4.3
Inferior and middle lobe	54	15.7
All superficial	24	6.9
Levels		
I	6	1.7
II	159	46.1
I and II	120	34.8
I to III	45	13
I to IV	9	2.6
III and IV	6	1.7
Histology		
Pleomorphic adenoma	153	44.3
Warthin tumor	132	38.3
Oncocytoma	15	4.3
First branch branchial cyst	15	4.3
Basal cell adenoma	9	2.6
Oncocytic papillary cystadenoma	9	2.6
Microcystic adenoma	6	1.8
Chondroma	3	0.9
Transient facial palsy		
Yes	84	24.3
No	261	75.7
Definitive facial palsy (branch)		
Yes	12	3.47
No	333	96.53
Facial nerve lesion by branch		
MMB	48	13.8
BB	6	1.8
ZB	9	2.6
OB	9	2.6
FB	9	2.6
All branches	15	4.3

MMB—marginal mandibular branch; BB—buccal branch; CB—zygomatic branch; OB—ocular branch; FB—frontal branch. Type of resection according to the ESGS: I = parotidectomy one level or extracapsular dissection; II = parotidectomy (one or two levels, more often partial superficial); III = parotidectomy (two levels, more often superficial); IV = parotidectomy (three or four levels removed, more often total). Anatomic situation in the gland corresponds to the clinical description by the surgeon; Levels according to ESGS classification: I = cranial superficial; II = caudal superficial; III = deep caudal; IV = deep cranial; V = accessory.

**Table 2 medsci-08-00042-t002:** Classification accuracy and performance results of the convolutional neural network (CNN) and multivariate logistic regression (MLR) on the testing set.

Model	Sensitivity (%)—95% CI	Specificity (%)—95% CI	Positive Predictive Value (%)—95% CI	Negative Predictive Value (%)—95% CI	Accuracy (%)—95% CI	ROC–AUC	95% CI—for the ROC–AUC
ANN	86.53 (79.41 to 91.47)	95.67 (85.30 to 99.14)	87.28 (83.16 to 92.23)	95.68 (86.13 to 98.19)	93.42 (88.34 to 96.19)	0.960	0.953 to 0.967
MLR	46.23 (41.11 to 51.18)	92.59 (86.33 to 94.87)	66.94 (61.27 to 69.17)	83.37 (78.47 to 87.29)	80.42 (76.16 to 83.11)	0.769	0.743 to 0.834

**Table 3 medsci-08-00042-t003:** Variable weight according to the logistic multivariant regression (LMR) and variable importance according to the artificial neural network (ANN).

LMR				ANN	
Variable	*p*	OR	95% Confidence Interval	Variable	Importance
Sex	0.004	0.409	0.221 to 0.755	Situation	0.227389
TOR	0.006	0.600	0.418 to 0.861	Vol CC	0.096018
Vol > 3 cm	0.517	1.423	0.490 to 4.134	Histology	0.030937
Situation	0.004	0.809	0.701 to 0.933	Vol AP	0.029990
Levels	0.156	0.829	0.640 to 1.074	Age	0.018491
TOI	0.788	1.096	0.562 to 2.136	TOI	0.017908
Histology	0.753	0.982	0.880 to 1.097	Levels	0.013154
Age	0.017	1.027	1.005 to 1.050	Vol < 3 cm	0.000694
Vol AP	0.434	0.803	0.464 to 1.391	TOR	−0.000495
Vol ML	0.075	1.624	0.953 to 2.769	Vol > 3 cm	−0.001494
Vol CC	0.953	1.013	0.662 to 1.550	Vol ML	−0.001810
				Sex	−0.022819

Abbreviations: TOR—type of resection; type of incision; AP—anteroposterior; ML—medial to lateral; CC—cephalocaudal.

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
