# Peer review of "Artificial Neural Network as a Tool to Predict Facial Nerve Palsy in Parotid Gland Surgery for Benign Tumors"

_medsci, 2020, doi:10.3390/medsci8040042_

Round 1

Reviewer 1 Report

The authors utilize machine learning with an artificial neural network to predict facial nerve dysfunction following parotidectomy. It is a common problem and has tremendous clinical significance.

The model utilizes intraoperative findings and is dependent on the surgical approach technique.

Thus it predicts what is clearly evident to the surgeon after the procedure is finished. How that adds to the management is unknown.

What would be important from a patient surgeon perspective would be if the model was able to identify preoperatively factors that could predict facial nerve weakness.

I think that this is an excellent pilot study and a start for something that may have clinical relevance in the future.

Reviewer 2 Report

Overall, the paper is interesting. I have a few comments:

  1. Please justify why use CNN on non-imaging data. CNN is designed to work with data for which the spatial/structural information matters.
  2. It is not a fair comparison to compare CNN with Logistic Regression, because the latter is a super simple predictive models, which is typically used to understand the linear relationships between the independent and dependent variables. Maybe the authors should compare CNN with SVM and random forest. 
  3. The limitation of small sample size should be discussed in Discussions section.

Author Response

Thanks for the reviewer for your precise observations. 

According to this suggestions, we make some modifications in our manuscript.

1.Please justify why use CNN on non-imaging data. CNN is designed to work with data for which the spatial/structural information matters.

Sorry for this mistake, as you can see the term convolutional corresponde to an error in the tittle page. But across the text we just use the term artificial neural network. Because, was the type of algorithms used. As you highlight, the use of CNN is for image analysis, and this is not the case, and this is not the algorithm. Thanks again for this appreciation. 

2.It is not a fair comparison to compare CNN with Logistic Regression, because the latter is a super simple predictive models, which is typically used to understand the linear relationships between the independent and dependent variables. Maybe the authors should compare CNN with SVM and random forest. 

We are agree, this is not a fare comparison, we extend in the discussion section why we compare just against LRM. Also, we are currently testing the algorithm against other ML algorithm like KNN, XgBoost or RF. But our results are not still available, and we hope to share this in a future publication. Moreover, we consider this type of comparison useful to present to other medical colleagues, because of represent a common statistical approach. 

3.The limitation of small sample size should be discussed in Discussions section.

We include this limitation in the discussion of our manuscript. 

Reviewer 3 Report

This is an interesting paper on the development of a model for predicting nerve damage at surgery in the parotid gland.

The manuscript must, however, go thorugh THOROUGH revision concerning the English language. 

Author Response

Thanks to the reviewer for his comment. 

We try to improve language mistake over the text. 

Round 2

Reviewer 2 Report

The revision looks good. It would be better to have a response letter to summarize the changes. Otherwise, it is hard to trace back my original review comments. 

Reviewer 3 Report

The language still needs PROFESSIONAL corrections.

For example line 260: "As a future perspective, comparison against other ML algorithms wants to be performed.....". That is not proper English.

There are more examples that MUST be corrected!